# Autoantibody binding and unique enzyme-substrate intermediate conformation of human transglutaminase 3

**Julie Elisabeth Heggelund** [1,2] ✉, **Saykat Das**[1,2], **Jorunn Stamnaes**[1,2], **Rasmus Iversen** [1,2] & **Ludvig M. Sollid** [1,2] ✉

Transglutaminase 3 (TG3), the autoantigen of dermatitis herpetiformis (DH), is a calcium dependent enzyme that targets glutamine residues in polypeptides for either transamidation or deamidation modifications. To become catalytically active TG3 requires proteolytic cleavage between the core domain and two C-terminal β-barrels (C1C2). Here, we report four X-ray crystal structures representing inactive and active conformations of human TG3 in complex with a TG3-specific Fab fragment of a DH patient derived antibody. We demonstrate that cleaved TG3, upon binding of a substrate-mimicking inhibitor, undergoes a large conformational change as a β-sheet in the catalytic core domain moves and C1C2 detaches. The unique enzyme-substrate conformation of TG3 without C1C2 is recognized by DH autoantibodies. The findings support a model where B-cell receptors of TG3-specific B cells bind and internalize TG3-gluten enzyme-substrate complexes thereby facilitating gluten-antigen presentation, T-cell help and autoantibody production.

Transglutaminase 3 (TG3) belongs to a family of calcium-dependent enzymes which in sequence-specific fashions target Gln residues in polypeptides for posttranslational modifications[1]. The reaction happens via a ping-pong mechanism, where a thioester bond involving active site Cys273 of TG3 and the side chain carbonyl of the substrate Gln is formed as an intermediate. The enzyme-substrate intermediate is resolved through a nucleophilic attack by a second substrate, which can either be a polyamine, the ε-NH$_2$ group of a polypeptide Lys residue or water. The two first reactions are termed transamidation, and when Lys is a substrate, it results in protein crosslinking by formation of an isopeptide bond. The latter reaction is termed deamidation and results in conversion of Gln to Glu.

TG3, like all human transglutaminase enzymes, has a four-domain structure with an N-terminal β-sandwich domain, two C-terminal β-barrel domains (C1C2) and a catalytic core domain that harbours a catalytic triad with Cys, His and Asp residues[1]. Several X-ray crystal structures have been determined for three human transglutaminases TG2, TG3 and factor XIIIA (FXIIIA) (Supplementary Table 1). Both TG2 and FXIIIA (PDB IDs TG2: 1KV3 and 2Q3Z; FXIIIA: 1GGU and 4KTY)[2–5]

were reported to undergo a large conformational change with displacement of C1C2 upon binding of an inhibitor mimicking a peptide substrate in the active site. TG3 is unique among transglutaminases in the requirement for proteolytic cleavage between the core domain and C1C2 to become catalytically active[6]. In the structure of cleaved TG3 with three bound calcium ions (PDB ID 1NUD)[7], C1C2 remains associated with the N-terminal and core domains through non-covalent interactions, and the active site Cys273 is engaged in a hydrogen bond with Tyr526 of the C1 domain. This Tyr residue is conserved among transglutaminases, and the Cys-Tyr hydrogen bond along with the occupation of the active site by a loop of the C1 domain is a shared feature in structures of transglutaminases with no substrate occupancy of the active site (PDB IDs human TG2: 1KV3, human TG3: 1L9M, human FXIIIA: 1GGU, and red sea bream transglutaminase: 1G0D)[2,4,8,9]. For TG2, the Tyr-Cys interaction was demonstrated to stabilize the closed, inactive enzyme conformation[10]. In a structure of FXIIIA with a bound irreversible peptidomimetic inhibitor, the Tyr residue undergoes large movement in conjunction with inhibitor binding and displacement of C1C2 (PDB ID 4KTY)[5]. Based on this structure of the

[1]KG Jebsen Coeliac Disease Research Centre, Institute of Clinical Medicine, University of Oslo, Oslo, Norway. [2]Department of Immunology, Oslo University Hospital-Rikshospitalet, Oslo, Norway. ✉e-mail: j.e.heggelund@medisin.uio.no; l.m.sollid@medisin.uio.no

enzyme-substrate intermediate of FXIIIA, it was hypothesized that a similar active TG3 conformation must exist, and atomic details of the calcium-binding sites of the active TG3 structure were predicted[5].

Transglutaminase activity is involved in the gluten-sensitive disorders celiac disease and dermatitis herpetiformis (DH) through creation of antigenic gluten epitopes by deamidation[11]. In addition, TG2 is a target of autoantibodies in celiac disease[12], whereas DH is characterized by autoantibodies to both TG2 and TG3[13]. Recently, we identified TG3-specific and TG2-specific plasma cells in duodenal biopsies of DH patients, and we generated a panel of TG3-specific monoclonal antibodies (mAbs) from single isolated cells[14]. These antibodies are specific to TG3 and do not cross-react to TG2. Further, we identified three distinct groups of conformational epitopes whose exact location within the TG3 structure remains unknown. These findings give credence to the notion that, via involvement of hapten-carrier like transglutaminase-gluten peptide complexes, TG3-specific B cells can receive help from gluten-specific T cells to become TG3-antibody producing plasma cells, much like TG2-specific B cells in celiac disease can receive help from gluten-specific T cells to become TG2-antibody producing plasma cells[15].

As demonstrated for TG2, such hapten-carrier-like complexes can take two different forms; they can be isopeptide-linked adducts where a gluten Gln residue is attached to a Lys residue on the enzyme surface, or they can be enzyme-substrate intermediates where the gluten peptide is bound in the active site[16]. The former is a stable product, whereas the latter is a transient state that will be generated as long as substrate is available. Although both forms can facilitate interactions between TG2-specific B cells and gluten-specific T cells[17], recent evidence points to the enzyme-substrate intermediate as the main driving antigen[18]. Unlike TG2, TG3 was observed not to catalyse formation of isopeptide bonds between gluten peptides and the enzyme itself[19], suggesting that interactions between TG3-specific B cells and gluten-specific T cells can only be fuelled by enzyme-substrate complexes. Thus, to be able to receive help from a gluten-specific T cell, a TG3-specific B cell must be able to bind TG3 with a bound gluten peptide substrate. It follows that it will be important to define the conformation of the enzyme-substrate intermediate of TG3 to understand the autoantibody production in DH.

Here, we describe both inactive and active conformations of human TG3 in complex with a TG3-specific Fab fragment derived from a DH patient. We show that cleaved TG3 with three bound calcium ions, upon binding of an inhibitor mimicking a substrate undergoes a large conformational change, as C1C2 including residue Tyr526 detaches from the rest of the enzyme. Importantly, the structure of the enzyme-substrate intermediate of TG3 without C1C2 was recognized by autoantibodies in DH patients, supporting a model where TG3-specific B cells internalize gluten peptides as part of TG3-gluten enzyme-substrate complexes. This will enable T-cell help by gluten-specific T cells and ultimately autoantibody production to TG3.

## Results
### Crystal structure of Fab DH63-B02 complexed with TG3
All structures of TG3 were solved in complex with the Fab fragment of the anti-TG3 antibody DH63-B02, derived from DH patient gut biopsies[14]. This Fab was chosen as it belongs to a group of DH autoantibodies which targets a major epitope region of TG3 (epitope group 2; N-terminally oriented). The observation that DH63-B02 does not recognize a TG3/TG2 chimeric molecule with the N-terminal domain from TG3 suggests that the antibody binds to the catalytic core rather than the N-terminal domain[14]. The initial structures were generated of zymogen TG3 (PDB ID 8OXV to 1.8 Å resolution) as well as dispase-cleaved TG3 with bound $Ca^{2+}$ (PDB ID 8OXW to 1.7 Å resolution) (Supplementary Table 2).

Both structures, being almost identical, contain three $Ca^{2+}$ bound at calcium-binding sites 1–3 as previously described in structures of

TG3[7,8] and FXIIIA[5,20]. CheckMyMetal[21] server analysis was consistent with binding of $Ca^{2+}$ and not any other metal ions (Supplementary Table 3). Moreover, replacing $Ca^{2+}$ with $Mg^{2+}$ in the highest resolution structure led to positive difference electron density, supporting the placement of $Ca^{2+}$ over $Mg^{2+}$ in all three positions. The fact that we find no evidence for binding of $Mg^{2+}$ is relevant, as replacement of $Ca^{2+}$ with $Mg^{2+}$ in binding site 3 was reported to cause structural changes and prevent the catalytic activity of TG3[7]. In both structures, the C1C2 domains are present with the Tyr526 residue of the C1 domain making a hydrogen bond with Cys273 thereby occluding the active site as described in previously published TG3 structures[7,8]. The loop (residues 461-473) containing the TG3 cleavage site is unresolved in both structures but was confirmed to be intact in the zymogen structure by analysing crystals by SDS-PAGE (Supplementary Fig. 1).

The structures further reveal that the Fab primarily binds to the catalytic domain of TG3 (Fig. 1a, b, Supplementary Tables 4 and 5). The paratope/epitope contacts are identical in the zymogen and the cleaved form of TG3. The Fab has direct H-bonds with residues of the catalytic domain of TG3 through all complementarity-determining region (CDR) loops of both the heavy (CDR-H) and light (CDR-L) chain except CDR-H2. The CDR loops also have several water-mediated interactions. The Fab makes slight interactions with the C2 domain by one residue of CDR-L2 and two residues in framework region 3 of the light chain. The epitope is located 13 Å away from the active site Cys and on the opposite side of the cleavable loop. Owing to this distance, it is unlikely that antibody binding affects the enzymatic function of TG3.

Our observation that the zymogen and the dispase-cleaved forms of TG3 are almost identical with Tyr526 engaged in hydrogen bonding to Cys273 in both structures, suggest that the enzyme must obtain a different conformation while acting on substrate. To investigate this possibility, we generated crystals of cleaved TG3 with an inhibitor bound in the active site as a proxy for substrate. We included Fab DH63-B02 and had $Ca^{2+}$ present in the crystallization solution.

The peptidomimetic inhibitor Z-DON (where "DON" refers to the Gln-mimicking electrophilic amino acid 6-diazo-5-oxo-L-norleucine) was chosen as active site inhibitor as it showed the highest activity for TG3 among several screened inhibitors[22]. Addition of Z-DON leads to irreversible alkylation of the active site Cys and thus captures TG3 in a conformation likely representing the substrate-bound intermediate. Formation of Z-DON adduct to Cys273 of TG3 was verified by mass spectrometry (Supplementary Fig. 2).

### Chromatography analysis of TG3 with and without Z-DON
Before undertaking structural studies, the effects of Z-DON binding to TG3 were assessed by size exclusion chromatography (SEC) analysis. Incubation with Z-DON changed the elution time of dispase-activated TG3 (from 14.5 to 16.2 ml), indicating a reduction in the overall size of the enzyme. SDS-PAGE analysis revealed that the C-terminal domains C1 and C2 were missing thus explaining the later elution (Fig. 2). Released C1C2 was not visible on the chromatogram, suggesting that it did not enter the column. The control sample that was treated equally, but not added Z-DON, kept the initial elution time. Evidence suggests that in vivo TG3 is cleaved by cathepsin L to generate an active enzyme[23]. To confirm that dispase- and cathepsin L-digested TG3 behave similarly, we repeated the experiment using cathepsin L-cleaved TG3. Also in this case, incubation with Z-DON induced detachment of the C1C2 fragment (Supplementary Fig. 3).

### Structural analysis of the catalytic site in the enzyme-substrate intermediate
The crystal structure of cleaved TG3 with Z-DON and $Ca^{2+}$ was solved at 2.5 Å resolution (PDB ID 8OXX, Supplementary Table 2). CheckMy-Metal analysis confirmed the occupancy of three metal binding sites by $Ca^{2+}$ (Supplementary Table 3). The data demonstrate that C1 and C2

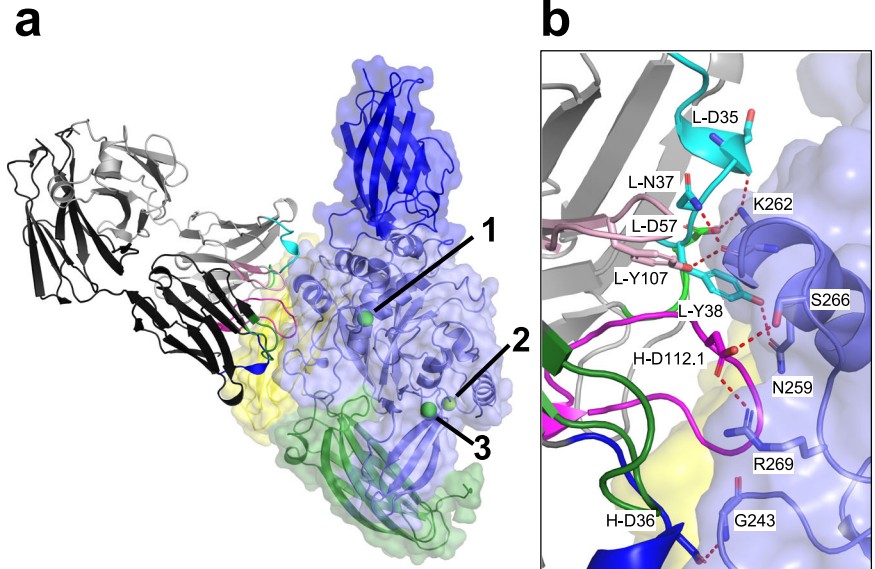

**Fig. 1 | Binding of Fab DH63-B02 to TG3. a** Structure of the complex of Fab DH63-B02 with TG3 (PDB ID 8OXW). The TG3 N-terminal domain is coloured dark blue, the catalytic domain is coloured light blue, the C1 domain is coloured green, and the C2 domain is coloured yellow. Ca$^{2+}$ ions are shown as green spheres, and labelled 1, 2 and 3 according to their binding site occupancies. The Fab heavy chain is coloured black, and the light chain is grey. The CDR1 loops are coloured blue (H) and cyan (L), the CDR2 loops are coloured dark green (H) and light green (L), and the CDR3 loops are coloured magenta (H) and light pink (L). **b** Closeup of paratope-epitope interactions with depiction of direct H-bonds (red broken lines) between residues of Fab DH63-B02 and TG3.

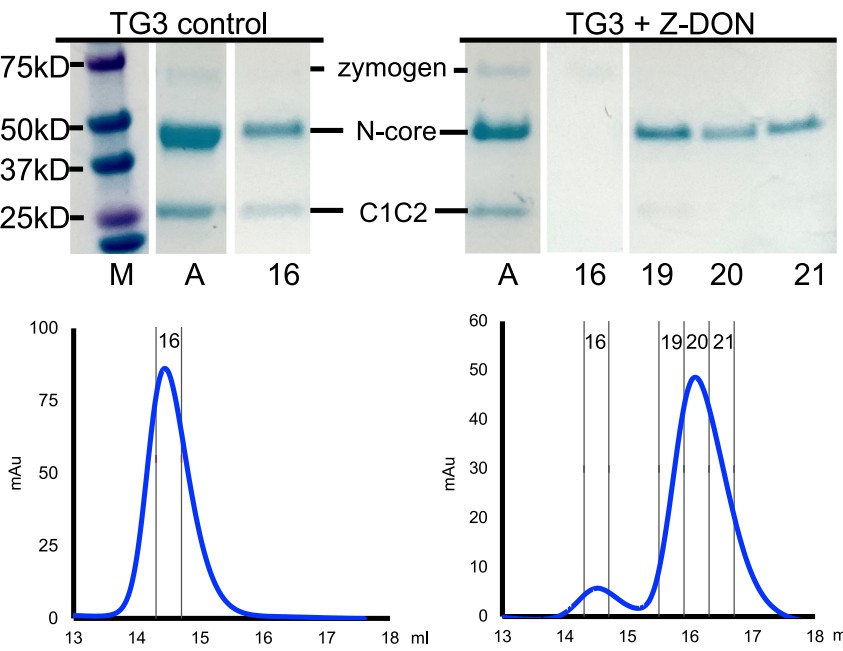

**Fig. 2 | Size exclusion chromatography (SEC) analysis of TG3 without or with Z-DON.** Top panels show SDS-PAGE analysis of the corresponding SEC fractions. M: Molecular weight marker. A: Sample applied to the column. 16–21. Fractions of the SEC run. Blue line: absorbance at 280 nm. The size of the zymogen is 79 kDa, the size of the N-terminal and core domains (N-core) of TG3 is 51 kDa, and the size of C1C2 is 26 kDa. Z-DON-incubated samples eluted in two peaks, where the smaller peak represented uncleaved (zymogen) TG3 as a result of incomplete dispase digestion. The experiment was repeated twice with similar results. Source data are provided in the Source data file.

indeed are missing from the structure (Fig. 3a). Thus, the binding of Z-DON leads to a structural rearrangement of TG3. In addition to the loss of the C-terminal domains, there is also a change in a β-sheet at the C-terminal end of the catalytic core domain, comprising residues 306–322 and 395–415 (Fig. 3b, Supplementary Movie 1). These sequences include some of the residues in the Ca$^{2+}$ binding sites 2 and 3

(see later). In the structures without Z-DON, the C1 domain is occluding the binding site, and Tyr526 is engaged in H-bonding with Cys273 (Fig. 3c, Supplementary Table 6). The detachment of the C1 domain facilitates the binding of Z-DON to Cys273, to Asn329 and to the backbone carbonyl of Trp237−residues that before detachment all bound to C1 (Fig. 3d−f, Supplementary Table 6). Four residues of the

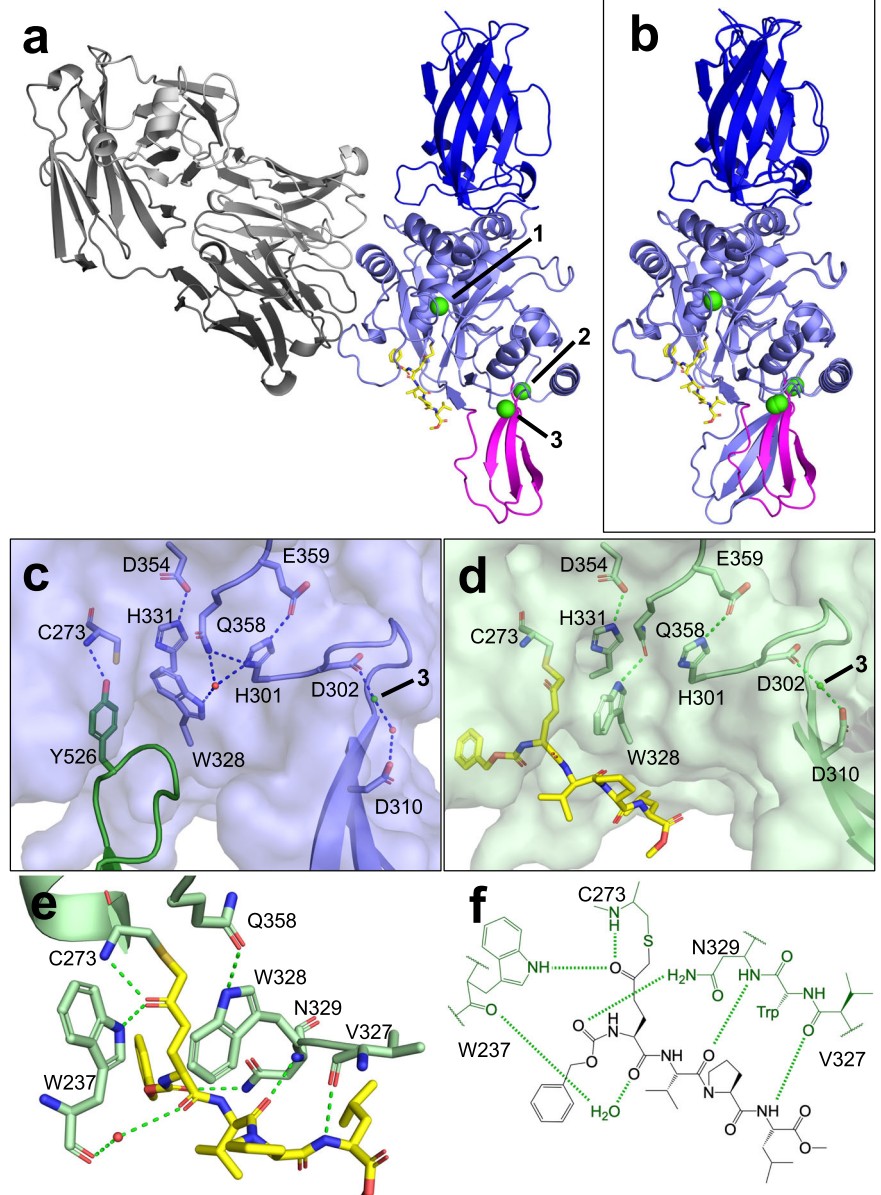

**Fig. 3 | Structure of TG3 with Z-DON inhibitor bound in the active site. a** The overall structure of cleaved TG3 + Z-DON + Fab DH63-B02 (PDB ID 8OXX). The β-sheet is highlighted in magenta, Z-DON is shown as yellow sticks, Ca²⁺ ions are shown as green spheres and labelled 1–3 according to their respective binding sites. **b** Superimposition of TG3 (N-terminal and core domain) with and without Z-DON, showing the movement of the β-sheet. **c** Closeup of the active site of TG3 without Z-DON, showing C1 in dark green (PDB ID 8OXW). The catalytic triad of Cys273, His331 and Asp354 are shown together with the proposed catalytic dyad His301 and Glu359. **d** Closeup of the active site of TG3 with Z-DON (yellow sticks) bound. **e** Closeup of Z-DON binding to TG3 with H-bonds shown in green and a water molecule shown in red. **f** Schematic representation of the interactions of TG3 with Z-DON. H-bonds are shown by green broken lines. TG3 residues engaged in binding of Z-DON are labelled.

β-sheet (Gly317, Asn318, Trp410 and Asn412) have their H-bonds to the C1 domain broken. The movement of the β-sheet enables Asp310 to directly co-ordinate Ca²⁺ at calcium-binding site 3, replacing a water molecule (Fig. 3c, d).

There are striking differences in the area around the active site of TG3 (Fig. 3c, d and Supplementary Movie 1). The side chain of Trp328 has turned by 90 degrees, exposing the active-site Cys273 by forming a tunnel that conceivably channels the Lys residue side chain to the acyl-enzyme thioester to exert its nucleophilic attack. Of note, this is not the same tunnel as described by Ahvazi et al.[7]. The movement of the Trp328 residue breaks its water-mediated contact with His301 and enables the formation of a direct H-bond between Trp328 and Gln358. His301 and Glu359 have been suggested to make up a catalytic dyad

facilitating the nucleophilic attack by Lys[5]. The interaction between these two residues is intact in all three structures with calcium. The arrangement of the active site as we observed in the inhibitor-bound conformation has not been seen in other published TG3 structures.

The active site of TG3 has a hydrophobic patch that is not seen in TG2 (Fig. 4). This patch is located at the N-terminal side of the inhibitor/substrate. This feature would select for hydrophobic residues at the N-terminal side of the substrate Gln and could explain that TG3, unlike TG2, was demonstrated to deamidate the sequence FPPQQPF (deamidation site underlined)[19]. Compared to sequences targeted by TG2, this sequence uniquely has a Pro in position −2 and a Phe in position −3. The restrained angles of Pro might cause a clash of Phe with hydrophilic TG2 residues Lys176 and Gln169, which are equivalent

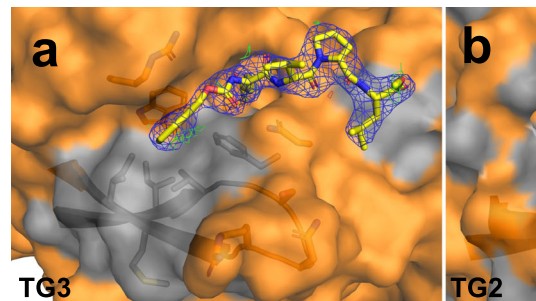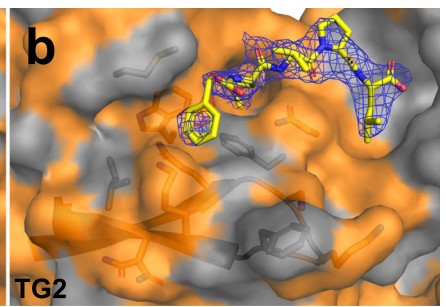

**Fig. 4 | Substrate binding sites of TG3 and TG2.** Hydrophobic residues are coloured grey, hydrophilic and polar residues are coloured orange. Electron density maps of Z-DON are generated in Coot for the TG3 structure and downloaded from the PDB for the TG2 structure. 2Fo-Fc maps are blue and contoured at 1.0σ, Fo-Fc maps are green/red and contoured at 3.0σ. **a** TG3 with Z-DON (PDB ID 8OXX). **b** TG2 with Z-DON (PDB ID 3S3J).

to the small hydrophobic residues Gly172 and Val165 in TG3. In structures of FXIIIA recently published, it was demonstrated how residues in the corresponding area of the substrate binding site (termed the α-space) have alternative conformations and move to accommodate different inhibitors/substrates[20].

### Details of the Ca²⁺ binding sites in TG3

The cleaved TG3 structure with Z-DON has several small but notable differences at Ca²⁺ binding site 3 compared to the TG3 structure without Z-DON. This site is co-ordinated by the side chains of Asp302, Asp304, Asn306, Asp310, Asp325, as well as the backbone of Ser308 (Fig. 5a–c). In the structure with Z-DON, the β-strand containing Asp310 is shifted 4 Å away from the Ca²⁺ and is instead binding to the ion via a water molecule. This structure of TG3 demonstrates a direct interaction between Asp310 and Ca²⁺, similarly to the inhibitor-bound FXIIIA structure[5]. The movement of Asp310 is part of a larger shift of the β-sheet in the C-terminal portion of the catalytic domain and widens the binding site of the inhibitor/substrate substantially.

The calcium ion at Ca²⁺ binding site 1 is co-ordinated by the side chains of Asn225 and Asn229, and the backbone carbonyls of Ala222, Asn225, Asn227 and a water molecule. These interactions are equivalent to the interactions in FXIIIA (called Ca²⁺ binding site 3 in FXIIIA)[5] and all published structures of TG3[7,8]. This site has been measured to have the highest affinity to Ca²⁺ in TG3[7].

At Ca²⁺ binding site 2 the calcium ion is co-ordinated by the side chains of Asn394, Glu444 and Glu449, the backbone carbonyl of Ser416 and two water molecules. These interactions are equivalent to the interactions in FXIIIA (called Ca²⁺ binding site 1 in FXIIIA)[5]. The Ca²⁺coordination at this site is identical in the structures with and without Z-DON.

### Structure of cleaved TG3 with Ca²⁺ removed

The crystal structures of TG2 are all without calcium ions. TG3 has been solved with one, two and three calcium ions, but never with zero ions (Supplementary Table 1). To investigate the effect of calcium on the structure of TG3, we added 10 mM EDTA to strip Ca²⁺ from cleaved TG3, mixed with Fab DH63-B02 as before, and grew crystals.

The structure was solved to 2.0 Å resolution and differs from the previously described structures in the three Ca²⁺ binding sites. Calcium-depleted cleaved TG3 has been analysed before[7], but never been crystallized and subjected to structural analysis. Changes at this site have effects on a loop bridging it to the active site of the enzyme (Fig. 6). Most notably, the His301 – Glu359 dyad is disrupted. This is enabled by the absence of Ca²⁺ in site 1 which frees Asn225 to form an H-bond with the carbonyl of Pro357, causing the Pro residue to flip by 180 degrees compared to the Ca²⁺ bound forms of TG3. This conformation constrains Gln358 and Glu359 to stay away from the active site of TG3. The residues Ala355 through Glu359 is predicted to be part of the binding site for the second (lysyl) substrate, and the large shift of

this region might explain why Ca²⁺ is crucial for this enzymatic reaction to occur.

In Ca²⁺ binding site 3, the lack of Ca²⁺ leads to a rearrangement of the loop containing Asp325 so that Ser324 can form an H-bond with Gly527 of the C1 domain. The loop is covering parts of the substrate binding site and has been described in previous structures that are also lacking Ca²⁺ in site 3 (Supplementary Table 1)[8]. Upon calcium binding, there is a loop movement which Ahvazi et al. described as the opening of a channel important for the catalytic function of TG3, possibly by controlling substrate access to the active site[7,8]. This channel is an opening between the core domain and the C1 domain, and we hence propose that this structural change rather is an integral part of the C1C2 domain detachment process. Other amino acid residues are kept in a similar position as in the structures with Ca²⁺, but many of the H-bonds between the catalytic domain and the C1C2 domain are tighter in the structure lacking calcium. Specifically, 10/17 H-bonds are tighter in the Ca²⁺-stripped enzyme compared to the cleaved TG3, while 14/17 are tighter in the Ca²⁺-stripped enzyme compared to the zymogen (Supplementary Table 6). Taken together, these observations suggest that the interactions between the catalytic domain and the C1C2 domains are weakened upon binding of Ca²⁺ and the enzymatic cleavage.

In Ca²⁺ binding site 2, the backbone keeps the same position in the two structures, while the side chains are positioned slightly differently. In the structure without Ca²⁺, Ser416 is peptide-plane flipped allowing the neighbouring His417 to form an H-bond to Glu449 (Fig. 6). In the TG2 structures without Ca²⁺ (PDB IDs 1KV3, 3LY6, 4PYG) the His417 equivalent Leu420 cannot form this H-bond, and Ser420 (corresponding to Ser416 in TG3), is interacting with the C1 domain of the closed conformation of TG2.

### Binding of DH mAbs to TG3 enzyme-substrate intermediate conformer

To test if TG3-specific antibodies in DH patients are reactive with the substrate-bound conformation of TG3, we used SEC-purified Z-DON-associated TG3 without C1C2 to assess binding of patient-derived mAbs in ELISA. Although the mAbs were originally isolated using the zymogen form of TG3[14], all 12 mAbs covering epitope groups 1, 2 and 3 retained good reactivity to the inhibitor-bound enzyme without the C1C2 domains (Fig. 7).

## Discussion

Here, we present the structure of TG3 with a peptidomimetic irreversible inhibitor bound at the active site. This structure likely represents the conformation of TG3 with the active site occupied by substrate. In this structure, as compared to the non-inhibitor bound structure, there are structural alterations of the catalytic core domain, and the two C-terminal β-barrel domains, C1 and C2, are missing. In the presence of Ca²⁺ and upon binding of substrate and formation of the thioester

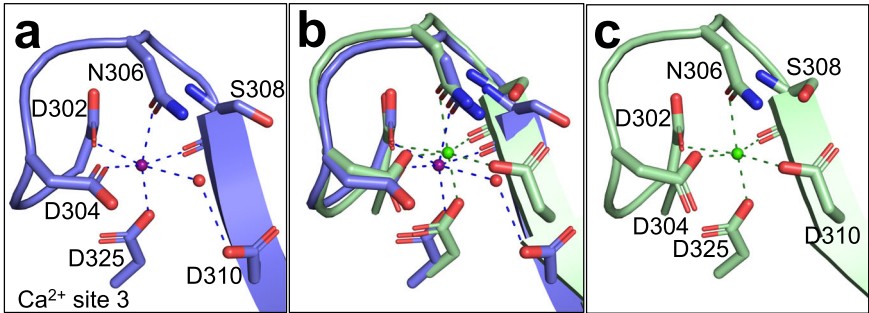

**Fig. 5 | Interactions at Ca²⁺ binding site 3.** Ca²⁺ is shown as either purple or green spheres, water molecules as red spheres and H-bonds as broken lines. **a** Closeup of Ca²⁺ binding site 3 in TG3 without Z-DON (PDB ID 8OXW). **b** Comparison of TG3 with and without Z-DON. **c** Closeup of Ca²⁺ binding site 3 in TG3 with Z-DON bound (PDB ID 8OXX).

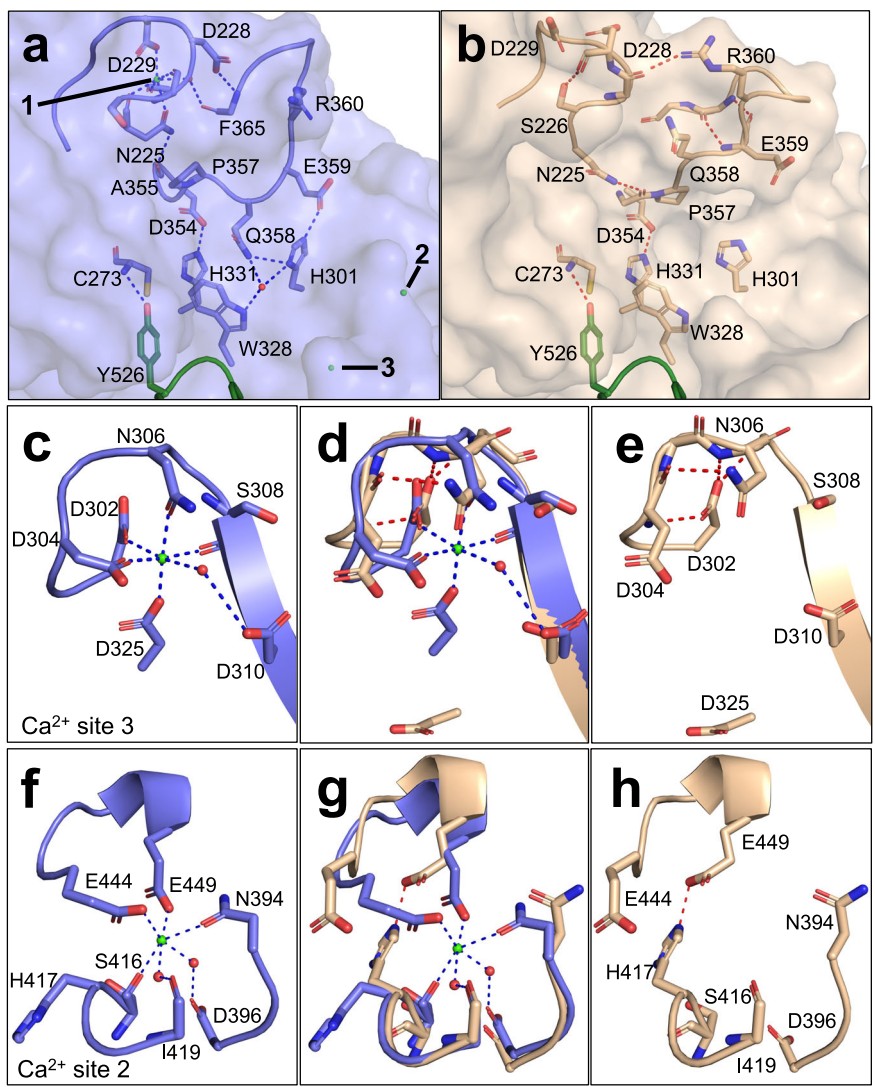

**Fig. 6 | The area around the Ca²⁺ binding sites in structures of TG3 with and without bound Ca²⁺.** The C1 domain with Tyr526 is shown in dark green. Ca²⁺ are shown as green spheres and are labelled 1–3, H-bonds are shown as broken lines. **a** Ca²⁺ binding site 1 and TG3 active site with Ca²⁺ (PDB ID 8OXW). **b** Ca²⁺ binding site 1 and TG3 active site without Ca²⁺ (PDB ID 8OXY). **c** Ca²⁺ binding site 3 with Ca²⁺. **d** Ca²⁺ binding site 3 with comparison between TG3 with and without Ca²⁺ bound. **e** Ca²⁺ binding site 3 in TG3 without Ca²⁺. **f** Ca²⁺ binding site 2 with Ca²⁺. **g** Ca²⁺ binding site 2 with comparison between TG3 with and without Ca²⁺ bound. **h** Ca²⁺ binding site 2 in TG3 without Ca²⁺.

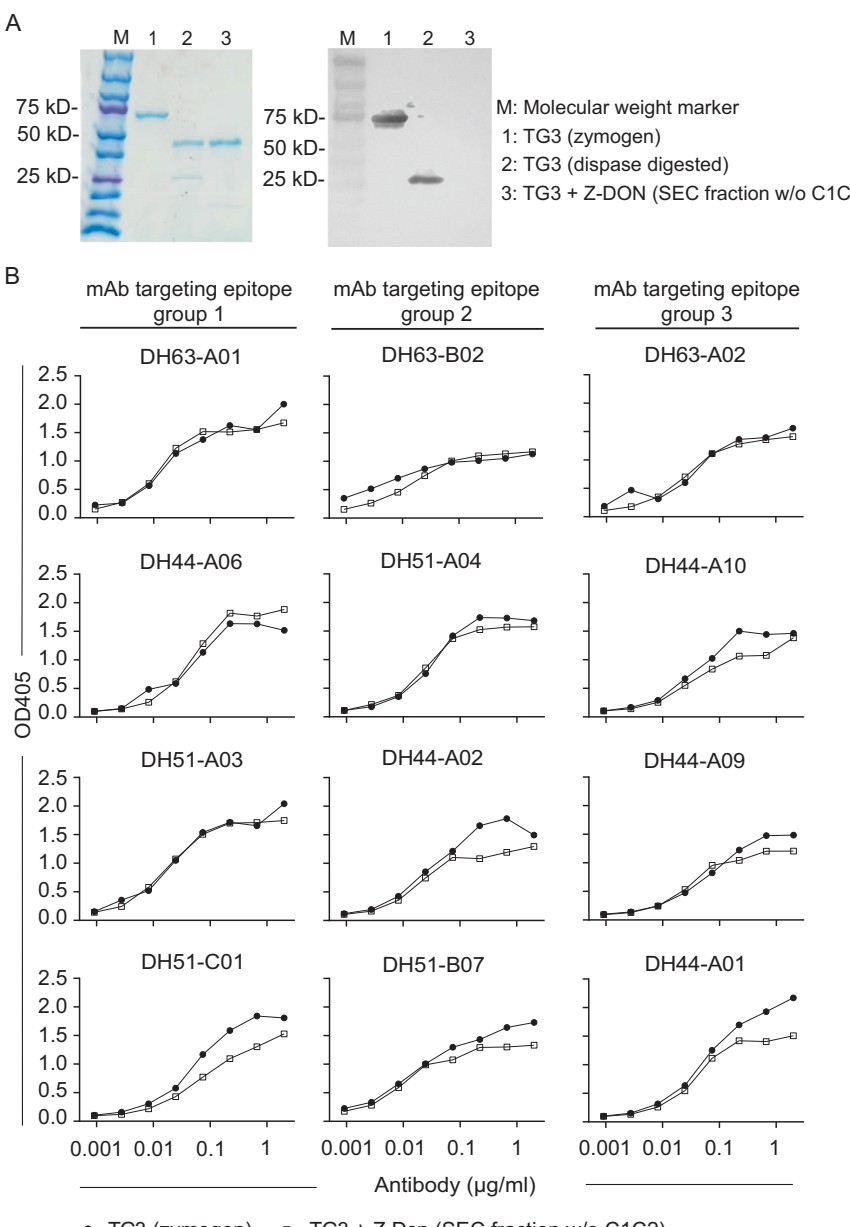

**Fig. 7 | Reactivity of DH patient derived mAbs to the enzyme-substrate intermediate conformer of TG3 without the C1C2 domains. A** SDS-PAGE analysis of different forms of TG3. The proteins were visualized by coomassie staining (left) or Western blotting (right) using a TG3-specific antibody that recognizes a C-terminal epitope. The protein loaded in lane 3 corresponds to SEC fraction 21 shown in Fig. 2. The analysis was repeated once with similar results. **B** ELISA binding curves showing reactivity of 12 TG3-specific mAbs generated from gut plasma cells of DH patients with the zymogen and enzyme-substrate conformers of TG3. The antibodies represent three distinct epitope groups as indicated. Source data are provided in the Source data file.

intermediate, TG3 will undergo a conformational change. The Cys273-Tyr526 hydrogen bond is broken, and in the active site the occupation by a loop of the C1 domain is replaced with the substrate (inhibitor). A tunnel for an incoming Lys is created to the active site allowing the ε-NH$_2$ group of Lys to exert its nucleophilic attack on the acyl-enzyme thioester, completing the catalytic cycle of the enzyme. A shift in a β-sheet located in the C-terminal part of TG3 impacts the interaction of the catalytic core domain with C1 and C2 making these two domains detach. The shift also changes the substrate binding site.

The two C-terminal domains have varying configuration in other inhibitor-bound transglutaminases; in FXIIIA (PDB ID 4KTY) they flip away from the active site to the other side of the protein[5], while in the TG2 structure (PDB ID 2Q3Z) they move so that TG2 adopts an extended, open conformation[3]. The latter structure does not contain any Ca$^{2+}$ which may indicate that it does not represent a catalytically active conformation of TG2. It is possible that the observed extended conformation of TG2 is a result of crystal contacts, as there are extensive interactions between C2 and an adjacent symmetry-related TG2 molecule in the crystal lattice. The catalytic dyad residues of TG2, His305 and Glu363, are not positioned in the same way in this TG2 structure as in the FXIIIA and TG3 structures. Interestingly, Alphafold v2.0 (alphafold.ebi.ac.uk) has modelled TG2 (P21980, [https://alphafold.ebi.ac.uk/entry/P21980]) to look more like the calcium-bound structures of TG3 and FXIIIA than all published TG2 structures. In the model the catalytic dyad is formed, and the residues of the Ca$^{2+}$ sites 1 and 3 are co-ordinated similarly to the corresponding sites in TG3 and FXIIIA. Biochemical analysis demonstrated that TG2 at increasing Ca$^{2+}$ concentrations can bind as many as

six calcium ions[24]. Five non-canonical $Ca^{2+}$ binding sites were defined by site-directed mutagenesis of which three correspond to the three $Ca^{2+}$ binding sites of TG3. Future studies are needed to identify the enzyme-substrate intermediate conformation of TG2 with bound calcium.

A common feature among transglutaminases may be a loosened interaction of C1 and C2 with the catalytic core domain upon substrate binding. In TG3, where cleavage of the linker between C1 and the catalytic core domain is required for activity, substrate binding appears to result in complete departure of the two domains. In other transglutaminases, where the linker is intact, likely due to different amino acid sequences in this region compared to TG3, C1 and C2 move away from the active site but remain physically attached to the rest of the protein. This may suggest that in the active conformation of these molecules, the C1 and C2 domains have functions that are not related to the catalytic activity of the enzyme. Of relevance to this notion, several functions have been ascribed to the C-terminal domains of TG2; the C1C2 domains interact with α2-macroglobulin in the pathway of low density lipoprotein receptor-related protein 1 (LRP1)-mediated uptake and endocytosis of TG2[25], they are responsible for homotypic association that can facilitate enzyme multimerization[26], and the C2 domain mediates binding of TG2 to an unknown extracellular matrix binding partner[27].

Here, we solve the structure of zymogen TG3 with three bound calcium ions. Comparison of this structure with that of cleaved TG3 with three bound calcium ions demonstrates that the two forms have identical conformations. Still, cleavage of the loop between the catalytic core domain and the C1 domain is required for the enzyme to become active. Hence, our findings suggest that rather than inducing conformational changes, cleavage enables the subsequent structural rearrangement that will take place upon binding of substrate/inhibitor.

As predicted previously[5], the catalytic dyad residues His301 and Glu359 likely are important for the transamidation reaction by TG3. Of note, in all our structures with calcium the two residues have the same arrangement. In FXIIIA, changing the His residue (His342) to Ala gave only a minor change in the affinity for the lysyl substrate, yet the catalytic transamidating activity was decreased by 85%[28]. In the structure of the calcium bound enzyme-substrate intermediate conformer of FXIIIA (PDB ID 4KTY), the two dyad residues (His 342 and Glu401), having an arrangement identical to that of our structures, were proposed to facilitate the nucleophilic attack by the Lys residue on the acyl-enzyme thioester. Further, Keillor et al. suggested that commonly for transglutaminases the His residue of the dyad serves as second base that mediates an initial deprotonation of the acyl-acceptor substrate prior to a second deprotonation by the active site general base during the nucleophilic attack[29]. Future analysis with mutation of the dyad residues and assessment of catalytic activity both when it comes to transamidation and deamidation should give further insight into the functional role of the dyad His and Glu residues in TG3 and other transglutaminases.

The underlying hypothesis for this work is that TG3-specific autoantibodies in DH are generated as a result of interactions between TG3-specific B cells and gluten-specific T cells facilitated by formation of TG3-gluten complexes, similar to what has previously been suggested for the TG2-specific autoantibodies in celiac disease[15]. Our observation that TG3-specific mAbs derived from DH patients retain good reactivity to substrate (inhibitor)-bound TG3 without the C1C2 domains speaks in favour of a model where the enzyme-substrate intermediate enables gluten antigen uptake by B cells, thereby facilitating the T cell-B cell interaction and production of TG3-specific autoantibodies. TG3-specific antibodies of DH patients recognize the TG3 enzyme both with and without bound substrate/inhibitor. In a situation where the enzyme is actively turning over substrate and cycles between peptide-bound and unbound states, a B cell likely must be able to recognize both conformations in order to bind TG3 stably and to efficiently internalize the intermediate via its B-cell receptor. This mechanism will allow uptake of gluten peptides by TG3-specific B cells followed by release of deamidated products in endosomes and subsequent presentation to T cells on HLA-DQ molecules.

Taken together, the structural analysis of complexes between TG3 and Fab of a DH patient autoantibody has given useful molecular insight into TG3 enzymology and autoantibody binding. The link between TG3 and DH has a striking parallel to the link between TG2 and celiac disease. Still, insight is missing about the active conformation of TG2 and where exactly the conformational epitopes of TG2 are located. Encouraged by the results of this study, future work should aim for similar analysis of TG2.

## Methods

### Production and purification of anti-TG3 antibody DH63-B02 Fab

DNA encoding the heavy and light chains of the anti-TG3 antibody DH63-B02 Fab were cloned into expression vectors and co-transfected into HEK 293-F cells (A14527, ThermoFisher Scientific) by using polyethyleneimine (23966-1, Polysciences). The transfected cells were cultured in shaker flasks at 37 °C, 8% $CO_2$ for 6 days before the supernatant was harvested. The secreted Fabs were purified by affinity chromatography. The supernatant was incubated with CaptureSelect LC-lambda (Hu) Affinity Matrix resin (084905, Life Technologies, Lot 110714-01) for 2 h before the slurry was applied to a gravity column. The resin was washed with 10 column volumes of wash buffer (sodium phosphate pH 7.2) and the protein was eluted with elution buffer (0.1 M glycine pH 2.5). The protein was neutralized by adding 0.2 ml neutralization buffer (0.1 M Tris pH 8.0) to each 1 ml of eluted protein. The purified protein was buffer exchanged into PBS and concentrated using 10,000 MWCO Vivaspin spin column (Cytiva) to approximately 3 mg/ml.

### Production and purification of human TG3

Human recombinant TG3 was produced in Sf9 insect cells as described[14]. Cell pellets were resuspended in lysis buffer: 50 mM sodium phosphate pH 8.0, 300 mM NaCl, 10 mM imidazole, 1% NP-40 substitute and EDTA-free cOmplete protease inhibitor cocktail (Roche) and incubated for 1 h on ice with vortexing every 15 min. The suspension was centrifuged at $18,000 \times g$ for 40 min and the supernatant was collected. His-tagged TG3 was purified from the lysate by Ni-NTA affinity chromatography. The resin was washed with 4 column volumes of lysis buffer without detergent and protease inhibitors. This wash was repeated twice to ensure a pure sample. TG3 was eluted using 3 column volumes of elution buffer (50 mM sodium phosphate pH 8.0, 300 mM NaCl and 250 mM imidazole) and dialysed into a storage buffer (20 mM Tris-HCl pH 7.4, 300 mM NaCl, 1 mM DTT, 1 mM EDTA). The purified protein was supplemented with 10% (v/v) glycerol and stored at −80 °C.

### SEC analysis of TG3

TG3 (zymogen) in storage buffer was cleaved by incubation with dispase I (D4818, Sigma) at 37 °C for 30 min, using 0.1 mg of dispase per mg of TG3 and supplemented with 5 mM $CaCl_2$. The dispase was subsequently removed by SEC using a Superdex 200 10/300 column (GE Healthcare) with running buffer 20 mM Tris pH 8, 150 mM NaCl, 8 mM $CaCl_2$, 1 mM EDTA, 1 mM DTT at a flow rate of 0.5 ml/min. The resulting peak fractions were combined and concentrated using a 10,000 molecular weight cut-off (MWCO) Vivaspin spin column (Cytiva). The irreversible active site inhibitor Z-DON (Z-DON-Val-Pro-Leu-OMe, Z006, Zedira) was added to the cleaved TG3 sample at a molar ratio of 25:1 and incubated at room temperature overnight together with a control sample without Z-DON. The next day, the samples were subjected to SEC as described above. The fractions of interest were individually concentrated using spin columns with 10,000 MWCO and analysed by SDS-PAGE using a 4–12% TGX gel (Bio-Rad) at 200 V for 30 min.

To analyse cathepsin L digestion of TG3, zymogen TG3 was diluted 1:1 in 0.1 M MES pH 6.0 and cleaved by incubation with cathepsin L (P07711, R&D Systems) at 37 °C for 30 min, using 0.003 mg cathepsin L per mg of TG3 and 10 mM $CaCl_2$. The sample was neutralized by the addition of Tris-HCl pH 9 to a concentration of 50 mM. Z-DON was added to the cathepsin L cleaved TG3 to a molar ratio of 120:1 and incubated at 4 °C overnight. A control sample without Z-DON was incubated in parallel. The next day, the samples were analysed by SEC using a Superdex 200 10/300 column (GE Healthcare) with TBS running buffer (50 mM Tris pH 7.6, 150 mM NaCl, 2 mM $CaCl_2$), at a flow rate of 0.5 ml/min.

## Western blot analysis

TG3 samples were loaded on SDS-PAGE gels (TGX gel, Bio-Rad) and blotted onto a nitrocellulose membrane (10600020, Cytiva) using a semi-dry transfer instrument (Bio-Rad). The membrane was incubated with polyclonal goat IgG specific for the C2 domain of TG3 (immunogen peptide residues 598–610) (PA5-37896, Invitrogen), diluted 1:3000 in TBS (50 mM Tris pH 7.6, 150 mM NaCl) with 0.1% (v/v) Tween-20. After incubation and washing, binding of the primary antibody was detected with peroxidase-conjugated donkey anti-goat IgG (705-035003, Jackson), diluted 1:7000 in TBS with 0.1% (v/v) Tween-20. Bands were visualized using enhanced chemiluminescence according to the manufacturer's instructions (SuperSignal WestPico Chemiluminescent Substrate, ThermoFisher Scientific), and recorded with a chemiluminescence reader (G:BOX Chemi XRQ, Syngene).

## ELISA

Zymogen TG3 or SEC purified TG3 without C1C2 were coated on polystyrene plates (Nunc Maxisorp Immuno-plate, ThermoFisher Scientific) at 1 µg/ml in TBS by overnight incubation at 4 °C. The wells were washed, in this step and subsequent washings, with TBS (50 mM Tris pH 7.6, 150 mM NaCl) supplemented with 0.1% (v/v) Tween-20 and 10 mM $CaCl_2$. Next, the wells were incubated with TG3-specific mAbs (expressed and purified as human IgG1[14]) diluted in TBS supplemented with 0.1% (v/v) Tween-20, 3% (w/v) BSA and 10 mM $CaCl_2$ for 1 h at room temperature. After washing, the wells were incubated with alkaline phosphatase-conjugated goat anti-human IgG (2040-04, Southern Biotech) diluted 1:2000 for 1 h at room temperature, washed again and incubated with 5 mg phosphatase substrate tablet (1003347284, ThermoFisher Scientific) dissolved in 10 ml phosphatase substrate buffer (1 M diethanolamine pH 9.8, 0.5 mM $MgCl_2$). Signals were measured in a plate reader (Multiskan ascent, ThermoFisher Scientific) at 405 nm.

## Mass spectrometry

TG3 (zymogen) and SEC-purified dispase-cleaved TG3 (2 µg, either unmodified or incubated with Z-DON and $CaCl_2$) was reduced with DTT and alkylated with iodoacetamide followed by precipitation on magnetic amine beads (MagReSyn, Resyn Biosciences)[30] for overnight digestion with sequencing grade trypsin (Promega). Tryptic digests were analysed by LC-MS/MS using the EvoSep One LC system (EvoSep Biosystems) coupled to an Orbitrap Q Exactive HF mass spectrometer (ThermoElectron, Bremen, Germany) equipped with a nanoelectrospray ion source (EasySpray/Thermo). For liquid chromatography separation we used a 15 cm C18 column (Dr Maisch C18 AQ, 3 µm beads, 100 µm ID, 15 cm long, EV-1074, with stainless steel emitter EV1086). The standard EVOSEP method 30 sample/day was used. The mass spectrometer was operated in the data-dependent mode to automatically switch between MS and MS/MS acquisition. Survey full scan MS spectra (from $m/z$ 375 to 1500) were acquired in the Orbitrap with resolution R = 60,000 at $m/z$ 200 (after accumulation to a target of 3,000,000 ions in the quadruple). The method used allowed sequential isolation of the most intense multiply charged ions, up to twenty, depending on signal intensity, for fragmentation on the HCD cell using

high-energy collision dissociation at a target value of 100,000 charges or maximum acquisition time of 15 ms. MS/MS scans were collected at 15,000 resolution at the Orbitrap cell. Target ions already selected for MS/MS were dynamically excluded for 30 seconds. General mass spectrometry conditions were: electrospray voltage, 1.9 kV; no sheath and auxiliary gas flow, heated capillary temperature of 250oC, normalized HCD collision energy 28%. Mass spectrometry raw files ($n = 3$, TG3 zymogen, SEC purified cleaved unmodified TG3, SEC purified cleaved Z-DON reacted TG3) were searched against a database containing the sequence of human TG3 with carbamidomethyl (+57 Da) or Z-DON (+600.3159 Da) as variable modification on cysteine (MaxQuant version 2.0.1.0)[31]. MS/MS spectra of peptides harbouring the active site Cys were extracted in Qual Browser (Thermo Xcalibur version 3.0.63, ThermoFisher Scientific) from raw files of SEC purified TG3 and verified by manual annotation of y-fragment ions.

## Preparation of complexes for crystallization

Zymogen TG3 was mixed with purified Fab DH63-B02 at a 1:1.1 molar ratio and incubated overnight at 4 °C. The complex was purified by SEC using a Superdex 200 10/300 column (GE Healthcare) with running buffer 20 mM Tris pH 8, 150 mM NaCl, 1 mM EDTA. The resulting peak fractions were pooled and concentrated to 17 mg/ml using a 10,000 MWCO Vivaspin spin column (Cytiva). We observed that not all TG3 was complexed with Fab using these conditions. To increase the fraction of Fab-bound TG3, all buffers were supplemented with $CaCl_2$ in subsequent experiments.

Zymogen TG3 was cleaved by incubation with dispase as described above. The dispase was removed by SEC as before but with 2 mM $CaCl_2$ supplemented in the running buffer (20 mM Tris pH 8, 150 mM NaCl, 1 mM EDTA, 2 mM $CaCl_2$). TG3 was subsequently mixed with Fab DH63-B02 at a 1:1.1 molar ratio and incubated overnight at 8 °C. The complex was purified by SEC, and the resulting peak fractions were pooled and concentrated to 7 mg/ml using a 10,000 MWCO Vivaspin spin column (Cytiva).

Cleaved TG3 was mixed with Z-DON at a 1:25 molar ratio, and with Fab DH63-B02 at a 1:1.1 molar ratio. The mixture was incubated at room temperature overnight and subsequently purified by SEC as before, but with 8 mM $CaCl_2$ supplemented in the running buffer (20 mM Tris pH 8, 150 mM NaCl, 1 mM EDTA, 2 mM $CaCl_2$). The resulting peak fractions were pooled and concentrated to 8 mg/ml using a 10,000 MWCO Vivaspin spin column (Cytiva).

To prepare the TG3 complex without calcium, 10 mM EDTA was added to cleaved TG3. After 10 min, Fab DH63-B02 was added to the sample at a 1:1.1 molar ratio and incubated overnight at 4 °C. The concentration of EDTA in the sample incubated overnight was 5 mM. The complex was purified by SEC as before, but with 1 mM EDTA and no $CaCl_2$ in the running buffer (20 mM Tris pH 8, 150 mM NaCl, 1 mM EDTA). Only ~10% of the protein was complexed, these fractions were pooled and concentrated to 5.1 mg/ml using a 10,000 MWCO Vivaspin spin column (Cytiva).

## Crystallization of TG3 complexes

The initial crystallization hits for the TG3 zymogen+Fab complex were obtained in the Morpheus crystallization screen (Molecular Dimensions, UK), row A containing 30 mM $CaCl_2$ and 30 mM $MgCl_2$, using a sitting-drop setup dispensed by a Mosquito robot (SPT Labtech, UK). These initial needle clusters were used to create microseeds by crushing the crystals with a seed bead (Hampton Research) according to the manufacturer's instructions. The microseeds were used in the following hanging-drop optimization experiments inspired by random microseed matrix screening, rMMS[32]. All three $CaCl_2$ complexes were crystallized in 0.1 M HEPES-MOPS pH 7.1, 16–22% ethylene glycol, 8–11% PEG 8000, 6–10 mM $CaCl_2$ and 3–10 mM $MgCl_2$. The TG3+Fab complex was cross-seeded with microseeds of TG3 zymogen+Fab, and the TG3+Fab+Z-DON complex was cross-seeded using microseeds of

TG3+Fab. Diffraction-quality crystals were obtained in 3–7 days. The crystal without $CaCl_2$ was grown in 0.1 M Bicine-Tris pH 8.5, 22% ethylene glycol and 11% PEG 8000, without microseeds, and took 2 months to grow.

For all setups, the crystals were cryo-protected using an additional 20% ethylene glycol and flash-cooled in liquid nitrogen.

### SDS-PAGE analysis of crystals

Single crystals were harvested from the same drop that yielded the crystal used to solve the structure of TG3 zymogen+Fab. The crystals were washed in crystallization well solution and dissolved in distilled water mixed with Laemmli loading buffer. The sample was boiled for 5 min before half of the sample was added β-mercaptoethanol. The samples were loaded onto a 4–12% TGX gel (Bio-Rad) and run at 200 V for 30 min.

### Data collection and refinement

Diffraction data were collected at the ESRF beam lines ID23-1, ID23-2, ID30A-3 and ID30B on four different occasions (25.02.2022, 31.05.2022, 22.09.2022 and 24.01.2023). The experimental sessions can be traced through the DOIs for 8OXV (10.15151/ESRF-ES-649173307), 8OXW (10.15151/ESRF-ES-771249081), 8OXX (10.15151/ESRF-ES-874802477) and 8OXY (10.15151/ESRF-ES-1022934233). The data collection and refinement statistics are shown in Supplementary Table 2. The data were indexed, scaled and merged by the ESRF autoprocessing software, before the unmerged and unscaled files were reprocessed using Aimless[33] in the CCP4 software suite[34]. All complexes crystallized in space group P2₁. The structures were solved using molecular replacement by the program Phaser MR[35]. The zymogen structure was solved using structures 1NUG[7] and an Alphafold Colabfold[36] model of the Fab DH63-B02 as molecular replacement search models. Parts of the Alphafold Fab model were not optimally placed in the structure and had to be rebuilt using Buccaneer[37] and manual rebuilding in Coot[38]. The TG3+Fab structure was solved using the refined TG3 zymogen+Fab structure as search model, and the TG3+Fab+Z-DON structure was solved using the refined TG3+Fab structure, with the C-terminal domains removed, as search model.

The structures were refined using REFMAC5[39] and manually optimized using Coot. Strong positive difference electron density was observed in the three $Ca^{2+}$ binding sites identified in previous TG3 structures, and in the active site for the TG3 + Z-DON structure. The Z-DON molecule and restraints library file were prepared using an isomeric SMILES string in Acedrg[40]. All four datasets were refined using translation/libration/screw-motion (TLS) restraints in REFMAC5. The TG3+Fab+Z-DON dataset was moderately anisotropic and was refined using external ProSmart restraints[41] for the Fab only. The loop connecting the catalytic core domain of TG3 to the C1C2 domains was not possible to place in any of the four structures due to weak electron density in the area and was consequently left out. The loops comprising residues 78–82 in TG3 and 136–142 in Fab heavy chain were difficult to place with confidence but were kept at their most likely positions. Metal ion binding was assessed using the server CheckMyMetal[21]. The final structures were validated using Molprobity[42], and by careful assessment of the PDB validation report. All figures were prepared with PyMol (Schrödinger LLC). Data refinement software versions are publicly available under each PDB entry experimental tab, and stated in the Reporting Summary.

### Reporting summary

Further information on research design is available in the Nature Portfolio Reporting Summary linked to this article.

## Data availability

The X-ray model co-ordinates and structure factors have been deposited in the Protein Data Bank (www.rcsb.org) under the accession codes 8OXV, 8OXW, 8OXX and 8OXY. Source data are provided with this paper.

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

## Acknowledgements

We acknowledge the European Synchrotron Radiation Facility (ESRF) for provision of synchrotron radiation facilities. The work was supported by grants from the South-Eastern Norway Regional Health Authority (project 2020027 to L.M.S.); Stiftelsen KG Jebsen (project SKGJ-MED-017 to L.M.S.) and the University of Oslo World-leading research programme on human immunology (WL-IMMUNOLOGY to L.M.S.) We would like to thank The Proteomics Core Facility at the Department of Immunology, University of Oslo/Oslo University Hospital for the mass spectrometry-based proteomic analyses. This facility is a member of the National Network of Advanced Proteomics Infrastructure (NAPI), which is funded by the Research Council of Norway INFRASTRUKTUR-program (project number: 295910).

## Author contributions

J.E.H.: Conceptualization, data curation, formal analysis, investigation, methodology, project administration, resources, validation, visualization, writing—original draft, writing—review and editing. S.D.: Data curation, formal analysis, investigation, formal analysis, resources, validation, visualization, writing—review and editing. J.S.: Data curation, formal analysis, investigation, visualization, writing—review and editing. R.I.: Formal analysis, investigation, resources, supervision, visualization, writing—original draft, writing—review and editing. L.M.S.: Conceptualization, formal analysis, funding acquisition, project administration, supervision, validation, visualization, writing—original draft, writing—review and editing.

## Competing interests

The authors declare no competing interests.
