## [Peer Review File · Nature Communications]

REVIEWER COMMENTS

Reviewer #1 (Remarks to the Author):

Ludvig Sollid and collaborators have recently isolated clonal antibodies from patients with Dermatitis Herpetiformis and shown that these fall into one of 3 epitope groups in terms of specificity. Here, a recombinantly produced Fab fragment is co-crystallized with TG3, the autoantigen recognized in these patients. Four x-ray structures are described corresponding to different physiologically relevant states of the enzyme. The main novelty is two-fold: Firstly, it defines a paratope/epitope interaction site of a patient derived antibody with the autoantigen and secondly, it provides the first structure of TG3 with a substrate-mimic in the active site. Importantly, the latter reveals cross-talk between substrate interaction and Ca²⁺-binding at site 3 which provides new insights into the catalytic mechanism. The binding epitope (termed group 2) recognized by the antibodies appears unperturbed by changes in enzyme configuration occurring during catalysis as indicated by similar apparent binding affinities for zymogen and enzyme irreversibly modified with substrate mimetic in the active site. There is now considerable evidence that interaction of IgD with the thioester intermediate of the enzyme with gluten peptides is what drives the autoantibody formation. The finding that the respective antibody epitope is preserved in different enzyme conformations is in line with this.

This is an excellent and rigorously conducted study that provides important new insights as outlined in paragraph above. The data is conclusive, experimentally well supported, and exceptionally well presented and discussed. Refinement statistics are indicating that high quality structural information has been obtained. However, there are a number of points that require further clarification as follows:

1. The authors have recently shown that group 2 epitope antibodies (ref 14) recognize a chimeric protein consisting of the N-terminal domain of TG3 fused to core/C1C2 domains of TG2. Given that the binding determinants identified in the crystal structure are primarily in the core domain (80XW), this finding is somewhat surprising and raises the questions whether analogous interactions could occur with the core domain of TG2 (although the antibodies have been shown not to bind native TG2).
2. Considerable efforts were made in previous work to remove Ca²⁺ from the purified TG3 zymogen. In fact, it turned out that the enzyme acquired the first Ca²⁺ ion during its synthesis (<https://doi.org/10.1093/emboj/21.9.2055>). Despite extensive efforts it was not possible to remove Ca²⁺ from site 1 (In fact, this work should be referred to at line 268/9 on page 12). Hence, it is somewhat surprising that simple EDTA treatment apparently removed Ca²⁺ from site 1 in the present work. The main difference appears to be that Ca²⁺ is stripped from a cleaved form of TG3 rather than the zymogen (a somewhat artificial scenario), and this therefore suggests that there are corresponding differences in the core domain in the loop consisting of Ile223-Val231 that connects Ca²⁺ site 1 and the enzyme active site between zymogen and Ca²⁺-stripped cleaved enzyme. Can a cleaved, Ca²⁺-stripped

and then reactivated enzyme (with Ca²⁺) gain back its full enzymatic activity? This aspect should be discussed in the manuscript.

3. It is also of note that complex formation of Fab with Ca²⁺-stripped cleaved enzyme is low as stated in methods (page 23, lines 523-525), perhaps suggesting reduced affinity? A more thorough analysis of binding kinetics for the different TG3 conformations may provide additional information, and perhaps reveal some additional insights that may not be apparent in equilibrium binding data.

specific comments:

In Fig 1, I suggest that authors number the three Ca²⁺-binding sites in order to provide the reader with the context for the detailed discussion in Figs. 5 & 6. Also, the choice of colour for the N-terminal domain and the core domain are a bit unfortunate as the two shades of blue are hard to distinguish.

In Fig 3, Asp 354 is incorrectly labeled as 'E354'. This should be corrected to 'D354'.

Page 10, lines 213-216 - relevant extended video data should be referred to.

Page 12, lines 292/3 - it should be made clear that this statement refers to a comparison between the zymogen and Ca²⁺-stripped cleaved enzyme.

Given that enzyme preparation is fundamental to these studies, it should be matched by the detail in which it is described. There is much information missing from the provided description: full details of buffers used in extraction and purification should be stated (page 20); how exactly was the enzyme concentrated for crystallization (page 22); were SEC conditions for all separations of complexes the same, i.e. as given on page 20?

Mg²⁺ at site 3: given the crystallization conditions, it is unlikely that Mg²⁺ occupancy would be observed at this site. The authors comment on this in the methods section (page 24, line 583-587). However, given that Ca²⁺-Mg²⁺ exchange has an essential role in regulating enzyme activity, I would suggest that a sentence is added in the results section to point this out to the reader and make this information more accessible.

Supplement:

Extended data videos - amino acid numbers should be given for ranges highlighted in colour (in legend)

Extended data table 1 - units are missing for refinement statistics (Rcryst/Rfree, percentage; rmsd bond length, angstrom; etc)

Reviewer #2 (Remarks to the Author):

Heggelund et al reported the several structures of TG2 in complex with autoantibody and found a large conformational change of beta-sheet in the catalytic core domain and C1C2 domain detaches. In addition, they found three Ca²⁺ binding sites, which is very controversial in the field of transglutaminase research field. Finally, they insist that their structural findings support a model where B-cell receptors of TG3-specific B cells bind and how its binding can facilitate gluten-antigen presentation and autoantibody production. I think this paper is a very interesting research result even though this paper contains limitations. Overall, the manuscript is well written and reflecting the main outcomes of the study. Regarding the new finding of TG3/autoantibody binding site, C1C2 processing, Ca²⁺ binding site, and et al, this paper can be recommended to be published in Nature communications after major revision as below;

1. Exact cleavage site of TG3??. Is the cleavage site conserved in other TG family?. Suggest to make the fig showing this fact.
2. How can the author so sure that their suggested Ca²⁺ binding sites contains Ca²⁺. Prove it experimentally. The ion characterization by electron density map is not proper way.
3. In the Fig2, can the author detect cleaved C1C2 fragment on SEC profile? I can not see it on profile at the right panel. Have to show it with making broaden retention volume. ~19 or 20??
4. In the fig3. beta sheet at the core domain was moved after binding of Z-DON. Is it similar with the case of TG2/Z-DON complex? Although C1C2 is not in the TG3/Z-DON complex, the comparison of structural movement of TG3/Z-DON and TG2/Z-DON will be interesting. Need comparison fig. at Fig3.
5. Fig4. Because this paper contains high resolution structure, it is worth to showing the electron density of Z-DON substrate and Ca²⁺ ion in the figs.
6. If the function of cleaved C1 and C2 of TG3 is not clear, it is good to analyse the activity of TG3 with or without cleavage. The effect of this cleavage on the transglutaminase activity of TG3.

POINT-BY-POINT RESPONSE, NCOMMS-23-27458-T

Reviewer #1 (Remarks to the Author):

Ludvig Sollid and collaborators have recently isolated clonal antibodies from patients with Dermatitis Herpetiformis and shown that these fall into one of 3 epitope groups in terms of specificity. Here, a recombinantly produced Fab fragment is co-crystallized with TG3, the autoantigen recognized in these patients. Four x-ray structures are described corresponding to different physiologically relevant states of the enzyme. The main novelty is two-fold: Firstly, it defines a paratope/epitope interaction site of a patient derived antibody with the autoantigen and secondly, it provides the first structure of TG3 with a substrate-mimic in the active site. Importantly, the latter reveals cross-talk between substrate interaction and Ca²⁺-binding at site 3 which provides new insights into the catalytic mechanism. The binding epitope (termed group 2) recognized by the antibodies appears unperturbed by changes in enzyme configuration occurring during catalysis as indicated by similar apparent binding affinities for zymogen and enzyme irreversibly modified with substrate mimetic in the active site. There is now considerable evidence that interaction of IgD with the thioester intermediate of the enzyme with gluten peptides is what drives the autoantibody formation. The finding that the respective antibody epitope is preserved in different enzyme conformations is in line with this.

This is an excellent and rigorously conducted study that provides important new insights as outlined in paragraph above. The data is conclusive, experimentally well supported, and exceptionally well presented and discussed. Refinement statistics are indicating that high quality structural information has been obtained. However, there are a number of points that require further clarification as follows:

RESPONSE: We thank the reviewer for the enthusiasm and the positive feedback to our work. This is much appreciated.

1. The authors have recently shown that group 2 epitope antibodies (ref 14) recognize a chimeric protein consisting of the N-terminal domain of TG3 fused to core/C1C2 domains of TG2. Given that the binding determinants identified in the crystal structure are primarily in the core domain (80XW), this finding is somewhat surprising and raises the questions whether analogous interactions could occur with the core domain of TG2 (although the antibodies have been shown not to bind native TG2).

RESPONSE: The reviewer is correct that in our previous study (Das et al., Adv Sci 2023, PMID: 37424036) we indicated that epitope group 2 is mainly located in the N-terminal domain. Hence it may seem paradoxical that Fab of DH63-B02 binds to the catalytic core domain. In our previous work (ref 14) we also showed that out of our 4 antibodies that belong to epitope group 2, DH63-B02 is the only one that does not bind to a chimeric TG3/TG2 molecule with the N-terminal domain from TG3 (see Fig 3D of the paper). This data suggest that this antibody binds to the catalytic core and not to the N-terminal domain, yet still it competes for TG3 binding with the three other epitope 2 group antibodies. More information about the previously observed binding of DH63-B02 to TG3 is now included in text.

2. Considerable efforts were made in previous work to remove Ca²⁺ from the purified TG3

zymogen. In fact, it turned out that the enzyme acquired the first Ca²⁺ ion during its synthesis (<https://doi.org/10.1093/emboj/21.9.2055>). Despite extensive efforts it was not possible to remove Ca²⁺ from site 1 (In fact, this work should be referred to at line 268/9 on page 12). Hence, it is somewhat surprising that simple EDTA treatment apparently removed Ca²⁺ from site 1 in the present work. The main difference appears to be that Ca²⁺ is stripped from a cleaved form of TG3 rather than the zymogen (a somewhat artificial scenario), and this therefore suggests that there are corresponding differences in the core domain in the loop consisting of Ile223-Val231 that connects Ca²⁺ site 1 and the enzyme active site between zymogen and Ca²⁺-stripped cleaved enzyme. Can a cleaved, Ca²⁺-stripped and then reactivated enzyme (with Ca²⁺) gain back its full enzymatic activity? This aspect should be discussed in the manuscript.

RESPONSE: We appreciate this valuable comment. The structure without calcium was obtained by including EDTA in the purification buffer, and EDTA was consequently also present in the crystallization drop solution. As is mentioned in the methods section, these crystals took much longer to grow (two months compared to the usual 1-2 weeks), so it is conceivable that it was the prolonged incubation that stripped the last calcium ion from the protein. We have reworded the text and made this information easier to find in the revised manuscript. We have also added the reference (formerly at line 268/9 on page 12) as suggested.

3. It is also of note that complex formation of Fab with Ca²⁺-stripped cleaved enzyme is low as stated in methods (page 23, lines 523-525), perhaps suggesting reduced affinity? A more thorough analysis of binding kinetics for the different TG3 conformations may provide additional information, and perhaps reveal some additional insights that may not be apparent in equilibrium binding data.

RESPONSE: We agree with the reviewer that this may indicate lower affinity of the Fab for the Ca²⁺-stripped enzyme. In the initial work even though we had this suspicion, we did not embark on examining the affinity of the Fab to the complex without Ca²⁺ for the mere reason that this represents an unphysiological setting. We will still argue that this is valid reasoning as *in vivo* TG3-specific antibodies bind TG3 in the extracellular environment where Ca²⁺ is present (1.1–1.4 mM for humans). For this reason, we prefer not to include such an analysis.

specific comments:

In Fig 1, I suggest that authors number the three Ca²⁺-binding sites in order to provide the reader with the context for the detailed discussion in Figs. 5 & 6. Also, the choice of colour for the N-terminal domain and the core domain are a bit unfortunate as the two shades of blue are hard to distinguish.

RESPONSE: We thank the reviewer for these excellent comments. The Ca²⁺ binding sites are now labeled as suggested, also in Fig 3a and in Fig 6a. In the previous version of Fig 1a, the surface coloring of the N-terminal domain was incorrect making its distinction from the core domain hard to see. This has now been rectified.

In Fig 3, Asp 354 is incorrectly labeled as 'E354'. This should be corrected to 'D354'.

RESPONSE: We thank the reviewer for spotting this error which is now corrected.

Page 10, lines 213-216 - relevant extended video data should be referred to.

RESPONSE: Done

Page 12, lines 292/3 - it should be made clear that this statement refers to a comparison between the zymogen and Ca²⁺-stripped cleaved enzyme.

RESPONSE: We thank the reviewer for pointing out this unclarity. Scrutinizing supplementary table 4, it is evident that 10/17 H-bonds are tighter in the Ca²⁺-stripped enzyme compared to the cleaved TG3, while 14/17 are tighter in the Ca²⁺-stripped enzyme compared to the zymogen. We have updated the text to make the statement clearer, and we have also corrected an erroneous number of H-bonds stated in the previous version.

Given that enzyme preparation is fundamental to these studies, it should be matched by the detail in which it is described. There is much information missing from the provided description: full details of buffers used in extraction and purification should be stated (page 20); how exactly was the enzyme concentrated for crystallization (page 22); were SEC conditions for all separations of complexes the same, i.e. as given on page 20?

RESPONSE: Further details on the methods are provided. We hope a sufficient degree of details is provided in the revised version of the manuscript.

Mg²⁺ at site 3: given the crystallization conditions, it is unlikely that Mg²⁺ occupancy would be observed at this site. The authors comment on this in the methods section (page 24, line 583-587). However, given that Ca²⁺-Mg²⁺ exchange has an essential role in regulating enzyme activity, I would suggest that a sentence is added in the results section to point this out to the reader and make this information more accessible.

RESPONSE: We thank the reviewer for this suggestion. The text describing the occupancy of metal binding sites has been moved to the Results section along with an addition of a table displaying the CheckMyMetal results (Supplementary table 2). As suggested, we have also added a comment on the reported impact on TG3 activity by exchanging Ca²⁺ with Mg²⁺ at binding site 3 (Ahvazi et al, J Biol Chem 2003, PMID: 12679341).

Supplement:

Extended data videos - amino acid numbers should be given for ranges highlighted in colour (in legend)

RESPONSE: Done

Extended data table 1 - units are missing for refinement statistics (R_{cryst}/R_{free}, percentage; rmsd bond length, angstrom; etc)

RESPONSE: The missing units have now been added.

Reviewer #2 (Remarks to the Author):

Heggelund et al reported the several structures of TG2 in complex with autoantibody and found a large conformational change of beta-sheet in the catalytic core domain and C1C2 domain detaches. In addition, they found three Ca²⁺ binding sites, which is very controversial in the field of transglutaminase research field. Finally, they insist that their structural findings support a model where B-cell receptors of TG3-specific B cells bind and how its binding can facilitate gluten-antigen presentation and autoantibody production. I think this paper is a very interesting research result even though this paper contains limitations. Overall, the manuscript is well written and reflecting the main outcomes of the study. Regarding the new finding of TG3/autoantibody binding site, C1C2 processing, Ca²⁺ binding site, and et al, this paper can be recommended to be published in Nature communications after major revision as below;

RESPONSE: We very much appreciate that the reviewer has found our work interesting.

1.Exact cleavage site of TG3??. Is the cleavage site conserved in other TG family?. Suggest to make the fig showing this fact.

RESPONSE: The exact cleavage sites by dispase and cathepsin L in TG3 have been determined previously. The figure below (taken from Cheng et al, JBC 2006, PMID: 16565075) illustrates the identified cleavage sites (dispase open triangle and cathepsin L closed triangle).

This sequence is not conserved among transglutaminases (see alignment below) which likely explains why the “TG3-zymogen activation mechanism” is not occurring for the other transglutaminase family members.

sp P00488 F13A_HUMAN	LIVTKQIGGDGMMDDITDITYKFQEGQEERLALLETALMYGAKKPLNTEGVM-----	513
sp P22735 TGM1_HUMAN	LIVTKAISSNMREDITYLYKHPEGSDAERKAVETAAAHGSKPNVYA--NR-----	573
sp P21980 TGM2_HUMAN	KISTKSVGRDEREDITHYKYPEGSSSEEREAFTRANHLNKLAEKE-----	469
sp O43548 TGM5_HUMAN	FISTKSIQSDERDDITENYKYEEGSLQERQVFLKALQ--KLKARSFHGSQRGAEIQPSRP	483
sp O95932 TGM6_HUMAN	CISTKAVGSDSRVDITDLYKYPEGSRKERQVYSKAVN--RLFGVEASGRRIWIRAG--G	478
sp Q08188 TGM3_HUMAN	YISTKAVGSNARMVDTKYKYPEGSDQERQVFKALG--KLK PNTPTFAAT -----	469
	* * * : : * : * * * . * * . *	
sp P00488 F13A_HUMAN	-----KSRSNVDMDFEV-ENAVLGKDFKLSITFRNNSHNRYTITAYL	554
sp P22735 TGM1_HUMAN	-----GSAEDVAMQVEA-QDAVMGQDLMVSVMLINHSSSRRTVKLHL	614
sp P21980 TGM2_HUMAN	-----ETGMAMRIRVQGSMMNGSDFVFAHITNNTAEEYVCRLLL	509
sp O43548 TGM5_HUMAN	TSLSQDSPRSLHTPSLRPSDVVQVSLKFKLLDPPNMGQDICFVLLALNMSSQFKDLKVN	543
sp O95932 TGM6_HUMAN	RCL---WRDDLLEP---ATKPSIAGKFKVLEPPMLGHDRLRLALCLANLTSRAQVRVNL	531
sp Q08188 TGM3_HUMAN	-----SSMGLTE-----EQEPSIIIGKLVAGMLAVGKEVNLVLLKLNLSRDTKTVTVM	519
	: . . : * : . . * :	

We have included text in the Discussion section that mentions the distinct sequence of TG3 in the cleavage region which likely explaining why it has this unique property among transglutaminases of zymogen cleavage and activation.

2. How can the author so sure that their suggested Ca²⁺ binding sites contains Ca²⁺. Prove it experimentally. The ion characterization by electron density map is not proper way.

RESPONSE: The metal binding sites of TG3 have been extensively characterized before done with equilibrium dialysis, inductively coupled plasma mass spectrometry and X-ray crystallography (Ahvazi et al, EMBO J 2002, PMID: 11980702; Ahvazi et al, JBC 2003, PMID: 12679341). This work demonstrated that TG3 has three metal binding sites that are occupied by Ca²⁺ or Mg²⁺. Binding site 1 and 2 can bind Ca²⁺ whereas binding site 3 can bind Ca²⁺ or Mg²⁺, but having a preference for binding of Ca²⁺. TG3 with two Ca²⁺ and one Mg²⁺ ions bound (PDB ID 1NUG) was found to be catalytically inactive (see response to reviewer #1 above), and to have a structure slightly different from a structure with three Ca²⁺ ions bound (PDB ID 1NUD). Importantly, in this structure the Mg²⁺ ion sitting in binding site three (by contrast to Ca²⁺ in this site) is unable to coordinate with residue D325 of TG3. Our two structures with Ca²⁺ but without inhibitor (PDB IDs 8OXV and 8OXW) match PDB ID 1NUD and not PDB ID 1NUG.

Our structures were generated in conditions with concentrations of Ca²⁺ dominating over Mg²⁺ further speaking to our structures containing Ca²⁺ at the metal binding sites.

The best evidence for presence of Ca²⁺ in our structures, however, comes from the direct scrutiny of our X-ray crystallography data. While analytical methods for detection on metal ions like atomic absorption spectroscopy and inductively coupled plasma mass spectrometry provide information about presence or absence of metal ions, X-ray crystallography provides information about the location of a metal within a macromolecule. X-ray crystallography has thus become a prominent method for assessing binding of metals to proteins (Kraus et al, Metallomics 2017, PMID: 28967006). Along with this development in the field, the accuracy of this type of analysis has become much better, not least because of tools like CheckMyMetal, (Zheng et al, Nat Protoc 2013, PMID: 24356774; Gucwa et al, Prot Sci 2023, PMID: 36464767) – the tool we used in our analysis. So, while we agree with the reviewer that X-ray crystallography may not have been considered proper for analysis metal binding to proteins, we will argue that this notion has changed.

CheckMyMetal (CMM) uses a validation algorithm for a systematic inspection of the metal-binding architectures in macromolecular structures. The algorithm is trained on high-resolution X-ray data (<1.5Å) and is continuously updated. It performs an in-depth analysis of the position, charge and type of atoms and residues surrounding the metal. The output is a detailed report listing the values of an ion site, highlighting discrepancies from the target values. The algorithm considers the characteristics of each metal ion site. For example, calcium is a divalent alkali-earth metal and is typically six-coordinated with octahedral geometry and a typical Ca-O distance of 2.4-2.5 Å. Calcium can also be coordinated by seven ligands, caused by at least one bidentate ligand. Calcium is usually coordinated by at least two carboxyl side chains from Asp or Glu, and often has water molecules in the first coordination sphere forming hydrogen bonds to a carboxyl side chain in the second

coordination sphere. We see all these characteristics in our TG3 structures. We have included the results of the CMM analysis in Supplementary Data Table 2. We would like to highlight the following features from this analysis:

Site 1 has octahedral geometry with one bidentate ligand (Asp229) and four other Ca-O bonds of appropriate length (2.37-2.45 Å). The water molecule is coordinated by the carboxyl side chain of Asp228 in the second coordination sphere. The valence is 2, which is the optimal value for an alkali-earth metal, and the r.m.s.d. from optimal geometry are within acceptable levels.

Site 2 has octahedral geometry with one bidentate ligand (Glu444) and one other carboxyl ligand (Glu449). In addition, this site has two other Ca-O bonds including two water molecules, all of appropriate values (2.33-2.46). One of the water molecules is coordinated by a carboxyl side chain (Asp396). A valency of 2.3 and a rmsd of 13.2 is within acceptable values supporting calcium.

Site 3 has octahedral geometry with three carboxyl side chain ligands and three additional Ca-O bonds including 1 water molecule. A valency of 2.3 and rmsd from optimal geometry are within acceptable levels. 8OXX has an additional carboxyl side chain instead of the water molecule.

We tried substituting magnesium into the structures. However, we got discrepancies for the bond lengths (too long for Mg-O preferred distance of 2.1 Å). Since the distance between Mg and O is so short, there is rarely room for several acidic coordinating ligands in the first shell.

Sodium is also an unlikely candidate since it rarely is coordinated by Asp or Glu side chains, and the valence would be too high (should be between 0.7 and 1.3 for sodium). Heavier atoms like Sr²⁺ or Ba²⁺ have longer metal-ligand distances and are also distinguishable by stronger electron density.

Based on previous work with TG3 and our CMM analysis we are confident that the three metal binding sites contain Ca²⁺ and not any other metal ions.

3. In the Fig2, can the author detect cleaved C1C2 fragment on SEC profile? I can not see it on profile at the right panel. Have to show it with making broaden retention volume. ~19 or 20??

RESPONSE: The reviewer is indeed right in that the C1C2 fragment is absent from the SEC profile. We have tried to find the eluted C1C2 without success, it is not present in any of the fractions. Our leading hypothesis is that the released C1C2 forms complexes with itself and is retained by the top filter of the column. Analysis of the homologous protein TG2 has indeed shown that it can form large non-covalent complexes when TG2 binds calcium and the C1C2 domains change conformation in relation to the core domain (Stamnaes et al, 2015, PMID: 26244572). We have observed a slight increase in the back pressure of the column over time. Since we do not have a definitive answer to this question, and it is likely not physiologically important for TG3, we did not include a discussion on this point in the manuscript.

4. In the fig3. beta sheet at the core domain was moved after binding of Z-DON. Is it similar with the case of TG2/Z-DON complex? Although C1C2 is not in the TG3/Z-DON complex, the

comparison of structural movement of TG3/Z-DON and TG2/Z-DON will be interesting. Need comparison fig. at Fig3.

RESPONSE: The TG2/Z-DON complex (PDB ID 3S3J) is without bound Ca^{2+} , so the comparison is not straightforward. Our results from TG3 demonstrate that binding of Ca^{2+} is an integral part of the structural rearrangement of the enzyme along with binding of the inhibitor. This essential feature is not represented in the PDB ID 3S3J structure. Thus, we will argue that the comparison is not very informative, and we prefer not to include it in the revised manuscript. We hope the reviewer will subscribe to our view.

5. Fig4. Because this paper contains high resolution structure, it is worth to showing the electron density of Z-DON substrate and Ca^{2+} ion in the figs.

RESPONSE: This is an excellent suggestion. Figure 4 is now updated with the electron density surrounding Z-DON in both panels.

6. If the function of cleaved C1 and C2 of TG3 is not clear, it is good to analyse the activity of TG3 with or without cleavage. The effect of this cleavage on the transglutaminase activity of TG3.

RESPONSE: We thank the reviewer for the comment. To the best of our knowledge, this issue has been settled already. As reported (Ahvazi et al, EMBO J 2002, PMID: 11980702) TG3 as a zymogen even in presence of Ca^{2+} is catalytically inactive.

REVIEWERS' COMMENTS

Reviewer #1 (Remarks to the Author):

The revised version of the manuscript by Heggelund et al in principal addresses the points raised by the reviewers, and the respective changes further improved this excellent manuscript. However, the answers to two of the points raised in the review process, although adequately addressed in the rebuttal, are not really "visible" in the manuscript. The respective information will be of interest to the specialist audience, and I suggest therefore that authors add a sentence for each to make this information more explicitly available.

reviewer 1, point 2:

Ca²⁺-stripping at binding site 1 - the conformation reported in structure 80XY may be dependent on cleavage between N/core- and C1/C2-domains (which would explain the discrepancy to the prior work, i.e. stripping of Ca²⁺ may be possible for cleaved enzyme but not zymogen). This possibility should be mentioned.

reviewer 2, point 3:

Instability of C1/C2 fragment following dissociation from N/core-domains - it should be mentioned that the C1/C2 fragment was not recoverable following dissociation.

Reviewer #2 (Remarks to the Author):

I am OK with their revision.

POINT-BY-POINT RESPONSE, NCOMMS-23-27458A

Reviewer #1 (Remark to the Author):

reviewer 1, point 2:

Ca²⁺-stripping at binding site 1 - the conformation reported in structure 80XY may be dependent on cleavage between N/core- and C1/C2-domains (which would explain the discrepancy to the prior work, i.e. stripping of Ca²⁺ may be possible for cleaved enzyme but not zymogen). This possibility should be mentioned.

RESPONSE: We have read again the papers by Ahvazi et al., (EMBO J 2002; PMID: 11980702 and J Biol Chem 2003; PMID: 12679341). In fact, while these authors in the 2002 paper reported that they were unable to strip Ca²⁺ from TG3, they reported in their 2003 paper that they indeed were able to remove 85% of Ca²⁺ from TG3 using 5 mM EGTA. Regrettably, this fact had missed our attention. The authors did not produce crystals and did not report the crystal structure of this protein. This is new in our work. Reassuringly, our results on Ca²⁺-stripping are consistent with the previous results. We have now changed the wording of our paper to accurately refer that TG3 stripped of Ca²⁺ has been reported before, but never crystallized (lines 257-258). We have also updated the methods with details of our Ca²⁺ stripping protocol (lines 530 + 532).

reviewer 2, point 3:

Instability of C1/C2 fragment following dissociation from N/core-domains - it should be mentioned that the C1/C2 fragment was not recoverable following dissociation.

RESPONSE: This fact is now mentioned in the revised text (lines 179-180).